# Exploring the effect of industrial agglomeration on income inequality in China

**Suhua Zhang** [ID]*, **Yasmin Bani** *, **Aslam Izah Selamat, Judhiana Abdul Ghani**

School of Business and Economics, Universiti Putra Malaysia, Kuala Lumpur, Malaysia

* zhangsuhua1023@gmail.com (SZ); nor_yasmin@upm.edu.my (YB)

## Abstract

Income inequality is a good indicator reflecting the quality of people's livelihood. There are many studies on the determinants of income inequality. However, few studies have been conducted on the impacts of industrial agglomeration on income inequality and their spatial correlation. The goal of this paper is to investigate the impact of China's industrial agglomeration on income inequality from a spatial perspective. Using data on China's 31 provinces from 2003 to 2020 and the spatial panel Durbin model, our results show that industrial agglomeration and income inequality present an inverted "U-shape" relationship, proving that they are the non-linear change. As the degree of industrial agglomeration increases, income inequality will rise, after it reaches a certain value, income inequality will drop. Therefore, Chinese government and enterprises had better pay attention to the spatial distribution of industrial agglomeration, thereby reducing China's regional income inequality.

## Introduction

Income inequality within a country has widened almost everywhere in the world over the past few decades. Income inequality has become one of the important issues within most of countries in the world today [1, 2]. Based on the World Inequality Report 2020, the richest 10% account for 75% of global wealth. Due to the rapid growth of emerging economies such as China, Brazil, India and Malaysia, inequality between countries has started to decline. However, the gap within countries has been increasing. Growing income inequality within country is viewed as one of the greatest social challenges. Because income inequality will be harmful to regional development and growth, foster great social and political instability [3], and matter to the well-being of individuals [4].

Over the past 40 years of reform and opening up, China's economy has achieved rapid development, but the problem of domestic income inequality has become more and more prominent. The Gini coefficient usually uses 0.4 as the "warning line" for income distribution gap. Based on China Statistical Yearbook (2021), China's Gini coefficient has exceeded 0.4 and the trend is constantly rising. This is extremely dangerous for China's stability. The Fifth Plenary Session of the 19th Central Committee of the Communist Party of China put forward long-term goals for 2035, including that all the people's common prosperity will achieve more obvious substantive progress in China, to actively address domestic income inequality. In order to achieve common prosperity, the Chinese government implemented the development

**Data Availability Statement:** All dataset files are available from the Dryad database (https://doi.org/10.5061/dryad.z08kprrht).

**Funding:** The authors received no specific funding for this work.

**Competing interests:** The authors have declared that no competing interests exist.

of the western region at the beginning of the 21st century, which encourages industrial agglomeration to move from the eastern region to the western region. In this regard, the study on whether industrial agglomeration has an impact on income inequality appears to be very important. If the answer is yes, to what extent does industrial agglomeration affect regional income inequality?

In recent years, some scholars have discussed the relationship between industrial agglomeration and income inequality. However, these is no universally accepted relationship between these variables. On the one hand, some scholars think that there is no significant correlation between industrial agglomeration and income inequality [5, 6]. On the other hand, some scholars believe that industrial agglomeration is the main cause of regional income inequality but there are different views on how industrial agglomeration affects income inequality. For example, Marchand [3] discovered that industrial agglomeration plays an important role in shaping income distribution and regions with high concentrations of manufacturing activities typically have lower income inequality. However, Xie et al. [7] believed that a high degree of industrial agglomeration will increase levels of inequality by attracting a large number of production factors to flow to the central area, promoting the rapid economic development of the central area, but causing relative poverty in the peripheral areas.

In light of the above opinions, this paper seeks to contribute to our understanding of the relationship between industrial agglomeration and income inequality in China from the following new perspectives. First of all, this paper presents an up-to-date portrait of the regional dimensions of income inequality across the country since the 21st century, which has been rarely depicted in the past. Secondly, most existing studies only focus on the linear impact of industrial agglomeration on income inequality, but the non-linear changes are not detailed enough. Industrial agglomeration affects income inequality through the agglomeration effect and crowding effect. It is easy to ignore the impact of the dynamic changes of the two effects on income inequality in the process of industrial agglomeration, which leads most studies to focus only on the linear relationship of the two variables. Therefore, this paper empirically investigates whether industrial agglomeration has a nonlinear relationship with income inequality. Finally, most previous studies have used traditional panel data analysis, ignoring the spatial correlation and spatial spillover effects of regional income inequality and industrial agglomeration. Thus, this paper uses global Moran's $I$ and local Moran's $I$ to test the spatial correlation of industrial agglomeration and income inequality respectively, then adopts the spatial Durbin model (SDM) to study the effect of industrial agglomeration on income inequality in China from 2003 to 2020.

The rest of this study is structured as follows. Section 2 reviews the literature on the link between industrial agglomeration and income inequality and other determinants of income inequality. Section 3 introduces the data, variables and methodology. Section 4 presents the results and discussion and in Section 5, the paper summarizes the suggestions and limitations.

## Literature review

### Measurement of industrial agglomeration

Industrial agglomeration has always been a hot topic in many disciplines. At present, there are many similar concepts to industrial agglomeration, such as enterprises agglomeration or cluster and industrial cluster. These three names are closely related concepts, but they have subtle differences. This paper mainly uses industrial agglomeration.

At the end of the 19th century, Marshall [8] proposed two important concepts, namely the "internal economy" and "external economy", which pay attention to the economic phenomenon of industrial agglomeration. The concepts of "industrial concentration zone" and

"Agglomeration Economies" were first proposed and used by Weber [9]. Porter [10] was the first to use "industrial agglomeration" to analyze cluster phenomena. Krugman [11] put forward that geographic agglomeration and specialization produce economies of scale, which in turn attract more companies to agglomerate and form industrial agglomerations. Han et al. [12] said that industrial agglomeration is defined as a cluster of companies in one or some interconnected industries concentrated in a certain area, which is united by common interests and complementary, and it is an important economic phenomenon in urban areas.

Based on the definition from Baidu Baike, industrial agglomeration refers to a process in which the same industry is highly concentrated in a certain geographic area, and the elements of industrial capital continue to converge within the space. To be specific, industrial agglomeration is a combination of many independent but interrelated enterprises and related supporting institutions that are highly concentrated in geographic space based on the relationship of specialized division of labor and collaboration, and form a strong and continuous competitive advantage.

Although there is consensus on the concept of industrial agglomeration, existing literature lacks a unified theory and method by which industrial agglomeration can be measured. Many scholars have devoted themselves to studying the measurement [13–15]. Of the many measurement methods, there are some commonly used methods to measure industrial agglomeration. [11] used spatial Gini coefficient to measure the degree of industrial agglomeration. Its value is between 0 and 1 and this method is widely used. However, its disadvantage is that the spatial Gini coefficient greater than 0 does not necessarily indicate the existence of clustering. Ellison and Glaeser [16] proposed a new agglomeration index, the Ellison-Glaeser Index (E-G index), to measure the geographic concentration of industries based on the spatial Gini coefficient. The larger the value, the more obvious the trend of industrial agglomeration. Duranton and Overman [17] estimated the distance density of a single variable by using the Gaussian kernel function, which is the Duranton-Overman Index (D-O index). Because the D-O index has very strict data requirements, it is difficult to study China's industrial agglomeration using the D-O index.

Although adopted by some scholars, the above methods are not suitable for this study because of the aforementioned shortcomings. Therefore, this paper employs the commonly used location quotient (LQ) to calculate the degree of industrial agglomeration. LQ is a very meaningful indicator to measure the spatial distribution of elements in a certain region and reflects the degree of specialization of a certain industrial sector. The greater the value of LQ, the greater the specialization rate. Munnich et al. [18] adopted LQ as the measurement standard. Peters [19] used LQ as the measurement standard. And he measured economic specialization for an industry in Missouri by calculating LQ for output, employment, compensation and foreign exports in 2000. Jiang and Xu [20] utilized location quotient (LQ) to measure the level of forestry industry agglomeration in Heilongjiang of China from the two perspectives of gross product and number of employees. Zhang et al. [21] employed LQ to measure the degree of industrial agglomeration taking industrial industries in different regions of China as research objects.

## Linkages between industrial agglomeration and income inequality

Studying the determinants of income inequality is a hotly debated topic in governmental policy and academic inquiry. However, the literatures on the relationship between industrial agglomeration and income inequality are few. There is a relationship between the two, which cannot be ignored [3, 22].

At present, the existing empirical literature on industrial agglomeration and income inequality mainly focuses on the regional and urban-rural levels. Regarding the regional level,

some empirical studies support that industrial agglomeration is not the main cause of regional income inequality. Meijers and Sandberg [6] and Maly [5] took Europe and the Czech Republic respectively as an example to empirically examine the inherent relationship between the multi-center agglomeration development model and the regional income inequality, and the results show that there is no significant correlation between the two.

On the other hand, some scholars believe that industrial agglomeration will affect the regional income inequality. Fan [23] empirically examined that regional specialization and manufacturing agglomeration are important reasons for the continuous expansion of the regional income inequality. Wang and Zhou [24] empirically proved that the impact of manufacturing agglomeration on regional income inequality has a significant inverted U-shape characteristic. At present, most regions in China are located to the left of the turning point, that is, the development of industrial agglomeration will still expand regional income inequality. Tao [25] empirically found that the effect of industrial agglomeration on the core and peripheral regions depends on the intensity of knowledge spillovers and economic growth. When knowledge spillovers are relatively weak, if industrial agglomeration promotes growth sufficiently, the underdeveloped regions will also obtain agglomeration dividends, thereby narrowing the regional gap.

Additionally, some scholars believe that the relationship between industrial agglomeration and regional income inequality has obvious regional heterogeneity. The empirical research of Xie et al. [7] shows that manufacturing agglomeration in China and the eastern region can help reduce the regional income inequality, while in the central and western regions it will widen the income inequality.

The above studies are analyzed from the regional level. In addition, some scholars analyze the effect of industrial agglomeration on the income inequality from the urban-rural level. Some empirical studies believe that industrial agglomeration has strengthened the dissemination of information in urban and rural areas, promoted the diffusion of advanced technologies [26], reduced transportation costs [27], stimulated the development of rural tourism [28], and created a large number of employment opportunities [29]. It can help rural areas develop better, thereby reducing the income inequality between urban and rural areas. Based on China's provincial panel data from 2005 to 2016, Peng and Yuan [30] used the dynamic spatial Durbin model (DSDM) to analyze the impact of industrial agglomeration on the urban-rural income inequality. They empirically tested that manufacturing agglomeration and construction industry agglomeration can narrow the urban-rural income inequality, and industrial agglomeration in coastal areas has played a significant role in narrowing the income inequality between urban and rural areas. Some scholars analyze the influence of the agglomeration of the circulation industry on the income inequality between urban and rural areas, and they also empirically discover that the higher the circulation industry agglomeration and the smaller the urban-rural income inequality [31, 32].

However, a few empirical studies hold the opposite view. They believe that the polarization effect of industrial agglomeration is in a dominant position in China [33], and various production factors in rural areas have obviously flowed to cities. When cities have been further developed, rural areas have fallen into production dilemma [34], the urban-rural income inequality will further widen.

There are currently few literatures on the relationship between industrial agglomeration and income inequality. Most studies on the determinants of income inequality focus on regional economic development level, human capital, government expenditure scale, and unemployment and so on. Regional economic development level and regional inequality show an inverted-U pattern or "Kuznets curve". Inequality will firstly increase as a country's economy rises and finally decrease due to structural change and equilibrium forces on capital and

labor [35]. Dunford and Perrons [36], Charron [37] and Iammarino et al. [38] empirically investigated that government expenditure scale is suggested to narrow the gap among regions and the governments should adopt heterogeneous development path of regions to deal with regional disparity. Behrens and Robert-Nicoud [39] and Castells-Quintana [40] empirically proved that regional inequality and population size are a U-shape relationship. They have tested that regional inequality tends to be highest in the cities which have the largest or smallest population. What's more, the majority of empirical studies proved that higher income inequality results from unemployment [41], and the age structure [22].

## Data and methodology

### Variables construction

This paper empirically analyzes the impact of industrial agglomeration on income inequality in China from 2003 to 2020. There are many indicators to measure the income inequality, the most commonly used are the Gini coefficient, the generalized entropy (GE coefficient) and the Theil index and so on. Internationally, the Gini coefficient is the most widely used, and its advantage is that it can objectively and intuitively reflect and monitor the income gap between residents, but it cannot be decomposed in groups. The GE coefficient examines the differences between individuals from the concepts of entropy and information. Better than the Gini coefficient, the GE coefficient can divide income gaps into intra-group gaps and inter-group gaps, but its calculation is cumbersome [42]. The Theil index is often used to analyze the urban-rural income gap [30], but it is not suitable for the inter-provincial income inequality studied in this paper. Therefore, based on [43], this study uses the proportion of the per capita income of each province to the national per capita income as the dependent variable to represent regional income inequality ($inequ_{it}$), which is a relative value not an absolute value.

The independent variable studied in this paper is manufacturing industrial agglomeration ($magg$). It adopts the location quotient (LQ) in calculations. The calculation formula is based on Zhang et al. [44] and Zhang and Bani [45]:

$$magg = LQ_{imagg} = \frac{M_{it}}{P_{it}} \frac{P_t}{M_t} \tag{1}$$

where $M_{it}$ is the manufacturing population of province $i$ at time $t$. $P_{it}$ is the total employment population of province $i$ at time $t$. $M_t$ and $P_t$ respectively represent the manufacturing population and total employment population of China at time $t$. Generally speaking, if LQ is bigger than 1, the manufacturing industry is highly agglomerated. If LQ is equal to 1, the degree of agglomeration of manufacturing industry is average. If LQ is less than 1, it indicates a low industrial agglomeration.

Table 1 shows the value of income inequality and manufacturing industrial agglomeration in 2020 and their growth rate in China, 2003 to 2020. The characteristics of dependent variable and independent variable can be summarized as follows. First, a high income inequality belt mainly focuses on the eastern coastal provinces in 2020 such as Beijing, Shanghai, Jiangsu, Zhejiang and Guangdong. The ratio of per capita income in the eastern region to the national per capita income is significantly higher than that in the western region, because eastern region has experienced fast development due to foreign direct investment (FDI) and its spillover effect since China's reform and opening up in 1978. Second, although regional income inequality still exists in China in 2020, based on Table 1, most eastern coastal provinces have negative growth rates in inequality, while many central and western provinces have positive growth rates. This is closely related to the great western development strategy in China at the beginning of the 21st century. Third, it is seen that manufacturing industrial agglomeration in

**Table 1. The income inequality and manufacturing industrial agglomeration in 2020 and their growth rate in China, 2003 to 2020.**

| Province | Income inequality | | Industrial agglomeration | |
|---|---|---|---|---|
| | Value | Growth rate | Value | Growth rate |
| Beijing | 2.0455 | -0.1592 | 0.3618 | -0.5276 |
| Tianjin | 1.2956 | -0.2638 | 1.1412 | -0.2322 |
| Hebei | 0.8474 | -0.0214 | 0.7861 | -0.1585 |
| Shanxi | 0.7884 | -0.0458 | 0.5859 | -0.2533 |
| Inner Mongolia | 0.9737 | 0.1089 | 0.5157 | -0.2386 |
| Liaoning | 0.9941 | -0.0716 | 0.9618 | -0.1351 |
| Jilin | 0.7874 | -0.1344 | 0.7894 | -0.1534 |
| Heilongjiang | 0.7597 | -0.1933 | 0.3966 | -0.5115 |
| Shanghai | 2.1056 | -0.1930 | 0.9050 | -0.3379 |
| Jiangsu | 1.3285 | 0.0068 | 1.5145 | 0.1105 |
| Zhejiang | 1.5833 | -0.1693 | 1.3503 | 0.2925 |
| Anhui | 0.8754 | 0.2090 | 1.0076 | 0.2214 |
| Fujian | 1.1391 | -0.1337 | 1.1979 | -0.2711 |
| Jiangxi | 0.8780 | 0.1045 | 1.0783 | 0.2919 |
| Shandong | 1.0090 | -0.0684 | 1.1142 | -0.1337 |
| Henan | 0.7733 | 0.0995 | 1.0007 | 0.1930 |
| Hubei | 0.8521 | -0.0764 | 0.9220 | -0.1560 |
| Hunan | 0.9166 | 0.0756 | 0.7154 | -0.0241 |
| Guangdong | 1.2422 | -0.3047 | 1.7587 | 0.3153 |
| Guangxi | 0.7669 | -0.0054 | 0.5583 | -0.2842 |
| Hainan | 0.8431 | -0.0895 | 0.2966 | -0.1488 |
| Chongqing | 0.9589 | 0.0235 | 0.8106 | -0.1572 |
| Sichuan | 0.8363 | 0.0820 | 0.7244 | -0.1668 |
| Guizhou | 0.7206 | 0.2831 | 0.3845 | -0.5140 |
| Yunnan | 0.7365 | 0.1216 | 0.5042 | -0.2607 |
| Tibet | 0.7045 | -0.1762 | 0.1862 | -0.0915 |
| Shaanxi | 0.8394 | 0.1992 | 0.6920 | -0.2991 |
| Gansu | 0.6612 | 0.0868 | 0.4836 | -0.4657 |
| Qinghai | 0.7680 | 0.0405 | 0.5810 | -0.0153 |
| Ningxia | 0.8210 | 0.1079 | 0.6027 | -0.1616 |
| Xinjiang | 0.7547 | -0.0209 | 0.4543 | 0.1222 |

the eastern region are significantly higher than in the central and west in China in 2020. Its positive growth rate mainly includes Guangdong, Jiangxi, Zhejiang, Anhui, Henan, Jiangsu and Xinjiang. It's because Xinjiang is the core area of China's "Silk Road Economic Belt", and the government is striving to improve Xinjiang's manufacturing capabilities. The positive growth in other provinces is mainly due to that industries tend to agglomerate in resource-rich areas and economically developed areas [12]. Finally, the growth rate, whether it is income inequality or industrial agglomeration, has seen the problem of spatial autocorrelation. Provinces experiencing rapid drops (increases) tend to be located near other provinces with similar drops (increases). In order to test this more precisely, next, this paper will calculate the Moran's $I$.

The source of data in Table 1 is the China Statistical Yearbook from 2004 to 2021.

The control variables of this study are as follows. Regional economic development level ($pergdp_{it}$) is the actual per capita GDP after the deflation by the consumer price index (CPI) in

**Table 2. Descriptive statistics.**

| Variable | Mean | Std.Dev | Minimum | Maximum | Source |
|---|---|---|---|---|---|
| *inequ* | 1.013 | 0.439 | 0.511 | 2.750 | CSY |
| *magg* | 0.835 | 0.362 | 0.0938 | 1.828 | CSY |
| *lnpergdp* | 10.11 | 0.649 | 8.190 | 11.59 | CSY |
| *unemp* | 3.490 | 0.701 | 1.210 | 6.500 | CSY |
| *lnGDR* | 3.609 | 0.193 | 2.959 | 4.057 | CSY |
| *human* | 10.83 | 1.479 | 4.516 | 16.83 | CLSY |
| *gov* | 0.247 | 0.187 | 0.0768 | 1.379 | CSY |
| *lnsize* | 8.096 | 0.855 | 5.599 | 9.443 | CPESY |

*Note*: CSY, CLSY and CPESY represent China Statistical Yearbook, China Labour Statistical Yearbook and China Population and Employment Statistics Yearbook respectively.

2003. Unemployment rate ($unemp_{it}$) is expressed by the ratio of unemployed persons to labor force in the region. Gross dependency ratio ($GDR_{it}$) refers to the ratio of non-working-age population to the working-age population, describing in general the number of non-working-age population that every 100 people at working ages will take care of. Human capital ($human_{it}$) is represented by the average years of education. Government expenditure scale ($gov_{it}$) is measured by the proportion of government fiscal expenditures in the region's GDP. Population size ($size_{it}$) is expressed by the number of people in each region. Because the control variables about $pergdp_{it}$, $GDR_{it}$ and $size_{it}$ vary greatly from 2003 to 2020 and their maximums and minimums are very different, the empirical studies will take the logarithm of these variables, such as $lnpergdp_{it}$, $lnGDR_{it}$ and $lnsize_{it}$.

The descriptive statistics with 558 observations for each variable in this paper are discussed in Table 2.

## Spatial correlation analysis

According to the previous content, there may be spatial autocorrelation between the dependent and independent variables in this study. If there is spatial autocorrelation of observations, spatial econometric models should be adopted to analyze the problem because ignoring the spatial autocorrelation can lead to biased results. Therefore, this section adopts Moran's $I$ to test their spatial autocorrelation. Helbich et al. [46] concluded that many scholars adopt index Moran's $I$ to test the spatial correlation of objects. There are two kinds of Moran's $I$, including the global spatial autocorrelation index and local spatial autocorrelation index. The formula of global spatial autocorrelation index Moran's $I$ is expressed as follows:

$$Global\ Morans\ I = \frac{n}{\sum_{i=1}^{n}\sum_{j=1}^{n}W_{ij}} \times \frac{\sum_{i=1}^{n}\sum_{j=1}^{n}W_{ij}(x_i - \bar{x})(x_j - \bar{x})}{\sum_{i=1}^{n}(x_i - \bar{x})^2} \qquad (2)$$

$$W_{ij} = \begin{cases} 1, & if\ province\ \text{i}\ and\ j\ are\ adjacent \\ 0, & if\ not \end{cases} \qquad (3)$$

where $n$ is the total number of provinces, $W_{ij}$ is the spatial matrix, $x_i$ and $x_j$ represent the observation in the $i$th province and $j$th province respectively, and $\bar{x}$ is the average of $x_i$ and $x_j$. The value of Moran's $I$ is between -1 and 1. When the value is equal to 0, this means no spatial relationship. If the value is less than 0, the correlation between samples is negative, which shows

**Table 3. Global Moran's *I* for income inequality and industrial agglomeration.**

| Years | Income inequality | | Industrial agglomeration | |
|---|---|---|---|---|
| | Moran's *I* | *p*-value | Moran's *I* | *p*-value |
| 2003 | 0.388 | 0.000 | 0.189 | 0.060 |
| 2004 | 0.388 | 0.000 | 0.233 | 0.023 |
| 2005 | 0.395 | 0.000 | 0.270 | 0.010 |
| 2006 | 0.407 | 0.000 | 0.288 | 0.006 |
| 2007 | 0.418 | 0.000 | 0.341 | 0.002 |
| 2008 | 0.428 | 0.000 | 0.344 | 0.001 |
| 2009 | 0.434 | 0.000 | 0.312 | 0.004 |
| 2010 | 0.449 | 0.000 | 0.318 | 0.003 |
| 2011 | 0.452 | 0.000 | 0.290 | 0.007 |
| 2012 | 0.451 | 0.000 | 0.308 | 0.004 |
| 2013 | 0.406 | 0.000 | 0.192 | 0.055 |
| 2014 | 0.406 | 0.000 | 0.181 | 0.068 |
| 2015 | 0.405 | 0.000 | 0.179 | 0.070 |
| 2016 | 0.406 | 0.000 | 0.194 | 0.053 |
| 2017 | 0.406 | 0.000 | 0.222 | 0.030 |
| 2018 | 0.403 | 0.000 | 0.235 | 0.022 |
| 2019 | 0.402 | 0.000 | 0.247 | 0.017 |
| 2020 | 0.402 | 0.000 | 0.260 | 0.013 |

that a negative spatial correlation exists in the variable. If the value is greater than 0, the correlation between samples is positive, which indicates that the variable represents a positive spatial correlation. When the value is closer to -1, the diffusion effect is stronger. On the other hand, the closer the value to 1, the more intense the agglomeration effect is. This paper conducts global space-related tests of income inequality and industrial agglomeration for 31 provinces in China from 2003 to 2020, analyzing the spatial interaction in income inequality or industrial agglomeration between provinces. The results are given in Table 3.

Table 3 shows the test results of global Moran's *I* of income inequality and industrial agglomeration in China from 2003 to 2020. Results show that the Moran's *I* values of income inequality are statistically significant at the 1% level and the values are positive. The Moran's *I* values of industrial agglomeration are also positive at a significant level of 5%, except in 2003, 2013, 2014, 2015 and 2016. The results illustrate that, at a significant level, China's income inequality or industrial agglomeration between provinces is not completely random. A spatial autocorrelation of income inequality or industrial agglomeration exists, which indicates the larger value is adjacent to the larger and the smaller value is adjacent to the smaller value in China.

In the above global correlation analysis, global Moran's *I* is significant, especially for income inequality. Therefore, income inequality is spatially correlated among the provinces in China. However, it is still unknown where the spatial agglomeration phenomenon exists. Therefore, this study uses the local Moran's *I* index to help explain the results further. Zhang et al. [44] put forward that the local Moran's *I* index is used to test the cluster-localized situation between observations. The formula for the local spatial autocorrelation index Moran's *I* is displayed as follows:

$$Local\ Morans\ I = \frac{(x_i - \bar{x})}{S^2} \sum_{j \neq 1}^{n} W_{ij}(x_i - \bar{x}) \tag{4}$$

A local Moran's *I* › 0 shows that a smaller value is surrounded by other small values (small —small), or a larger value is surrounded by other large values (large—large). What's more, Moran's I ‹ 0 indicates that a larger value is surrounded by small values (large–small), or a smaller value is surrounded by large values (small–large).

This paper uses Moran scatterplots to further verify the spatial correlation between income inequality and manufacturing industrial agglomeration. Figs 1 and 2 present the Moran scatterplots of income inequality and industrial agglomeration for 31 Chinese provinces in 2003, 2009, 2014 and 2020 respectively. The sample value of 31 provinces is not randomly distributed in four quadrants but rather in a regular gathering distribution. Regarding the distribution of Moran scatterplots of income inequality, there are 24 provinces located in the first and third quadrant in 2003, 2009 and 2020, and 22 in 2014, where income inequality is low and the surrounding provinces' inequalities are also low, such as Guizhou, Yunnan and Tibet, and those provinces with higher income inequality have neighboring provinces with higher income inequality, such as Fujian, Tianjin and Beijing. Additionally, the number of the provinces in the first and third quadrants accounts for about 77% in these four years. For the independent variable industrial agglomeration, the numbers of scattered points from 2003 to 2020 located in the first and third quadrants are 18, 20, 23 and 21 respectively. Compared with the Moran's

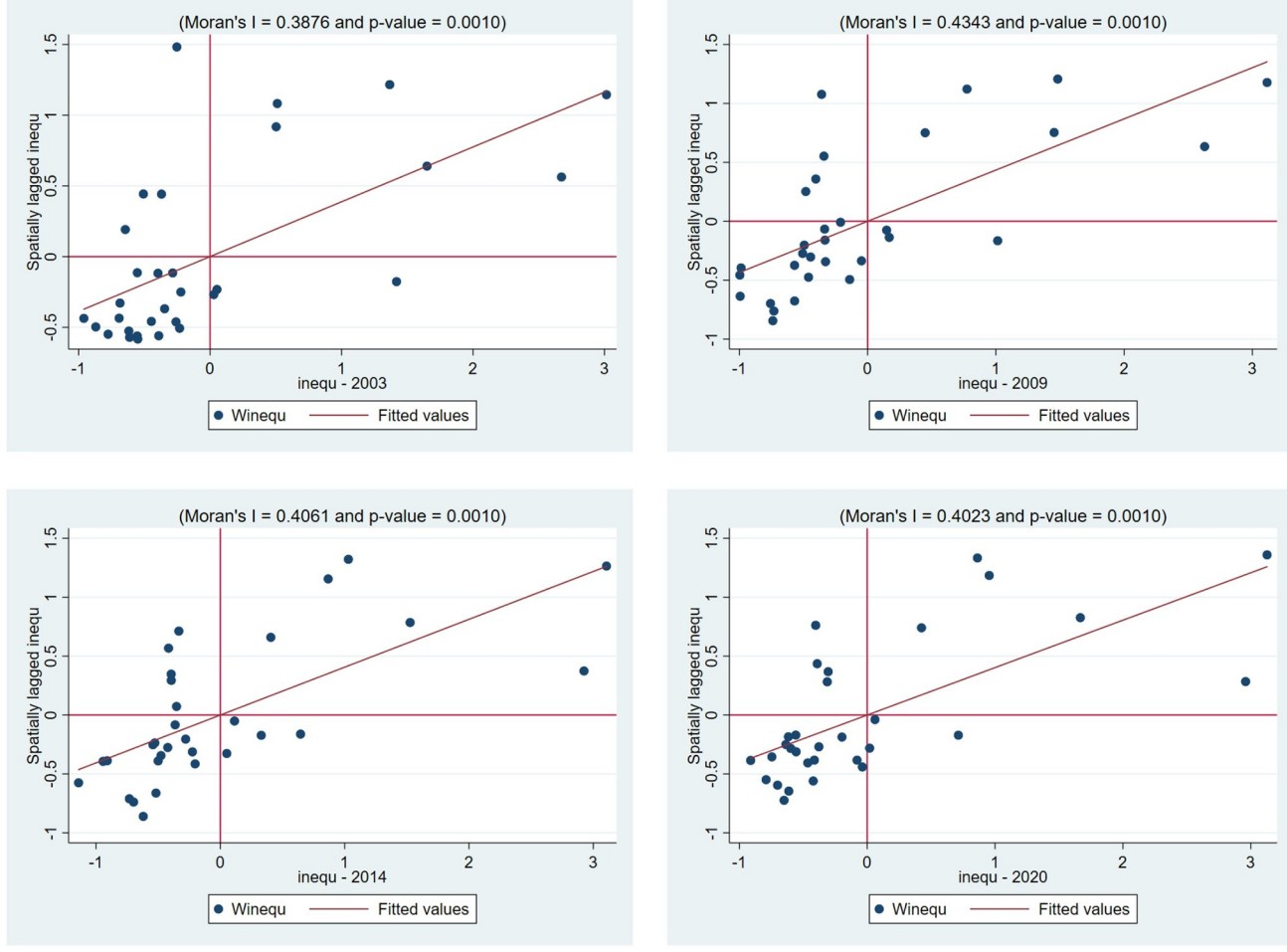

**Fig 1. Moran scatterplots of income inequality in 2003, 2009, 2014, and 2020.**

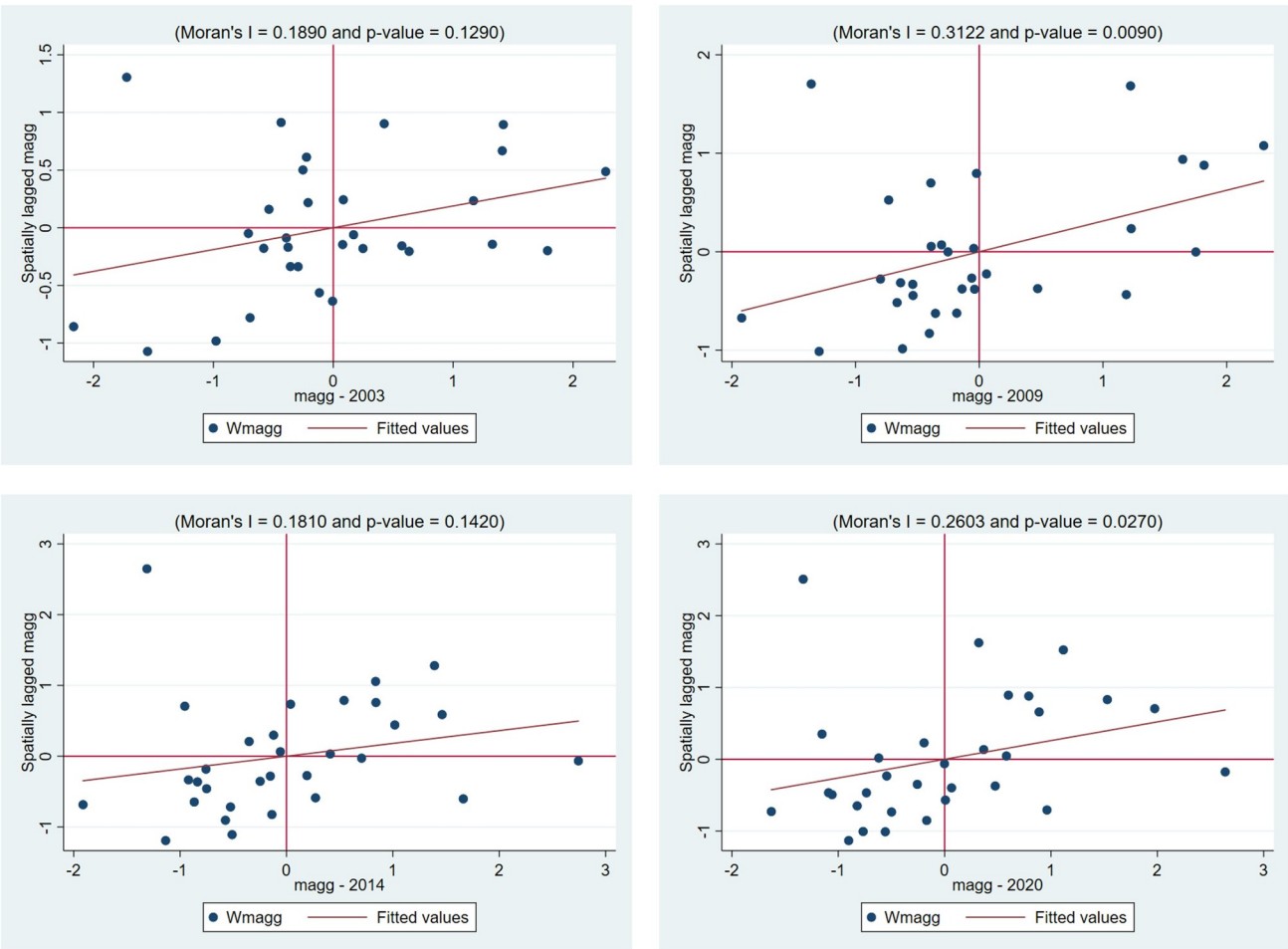

**Fig 2. Moran scatterplots of manufacturing industrial agglomeration in 2003, 2009, 2014, and 2020.**

*I* values of income inequality, the Moran's *I* values of industrial agglomeration in recent years are not very big, but it can not be ignored either. Thus, Figs 1 and 2 again confirm that significant spatial autocorrelation of income inequality exists.

## Model specification

The initial OLS model is based on Xie et al. [7] and Wang and Zhou [24].

$$inequ_{it} = al_{it} + \beta_1 magg_{it} + \beta_2 magg_{it}^2 + \sum \delta \times CV_{it} + \varepsilon_{it} \tag{5}$$

where $i$ and $t$ represent the province and time period respectively. The dependent variable $inequ_{it}$ is income inequality. $l_{it}$ is a vector of constant terms. $magg_{it}$ is manufacturing industrial agglomeration. The independent variables are $magg_{it}$ and $magg_{it}^2$. According to Petrakos et al. [47], Cuaresma et al. [48], Breau [22], Essletzbichler [41], Charron [37], Jiang and Kim [49], Castells-Quintana [40] and Iammarino et al. [38], $CV_{it}$ is a series of control variables, including *lnpergdp*, *unemp*, *lnGDR*, *human*, *gov* and *lnsize*. $\alpha$, $\beta_1$ and $\beta_2$ are the coefficients and $\varepsilon_{it}$ is the error term.

This paper studies the industrial agglomeration that not only affects income inequality in the province, but also income inequality of surrounding provinces. Income inequality between neighboring provinces also has spatial correlation and spatial spillover effects. Therefore, this paper will adopt a spatial panel model, the spatial Durbin model (SDM), which can take into account the spatial dependence of the dependent variable and the independent variables at the same time. The spatial econometric model was first proposed by Cliff and Ord [50], initially aimed at cross-sectional data and then expanded into a panel model by Anselin [51], Lee and Yu [52] and Zhao et al. [53]:

$$inequ_{it} = \rho \sum_{j=1}^{n} W_{ij} inequ_{it} + al_{it} + \beta_1 magg_{it} + \beta_2 magg_{it}^2 +$$
$$\theta \sum_{j=1}^{n} W_{ij} magg_{it} + \sum \delta \times CV_{it} + \varphi \sum_{j=1}^{n} W_{ij} CV_{it} + \varepsilon_{it}$$
(6)

where $W_{ij}$ is the spatial weight matrix, which sets the weight matrix of 0 and 1 according to whether the space between the two regions is adjacent. $\rho$, $\theta$ and $\varphi$ are the spatial coefficients. For instance, $\rho \sum_{j=1}^{n} W_{ij} inequ_{jt}$ is the interactive relationship between the dependent variables in adjacent regions. If $\rho > 0$, there is a spatial spillover effect of the dependent variable in the neighboring area. If $\rho < 0$, there is a siphon effect in the neighboring area–that is, the region with stronger economic strength–and development potential attracts the superior resources from the neighboring region.

$$W_{ij} = \begin{cases} 1, & if \ province \ \text{i} \ and \ j \ are \ adjacent \\ 0, & if \ not \end{cases}$$
(7)

Additionally, a combination of LM_Error, RLM_Error (spatial error robustness test), LM_Lag and RLM_Lag (spatial lag robustness test) is adopted to further validate why this study uses the spatial Durbin model (SDM) rather than other common spatial models. The results are shown in Table 4. Based on the model without spatial effect (except LM_Lag), all the null hypotheses are rejected. Thus, SDM model is usually given priority because both the spatial autoregressive model (SAR) and the spatial errors model (SEM) can be accepted [54]. This paper also uses LR test and Wald test to prove whether SDM model can be degenerated into SAR model or SEM model. It is obvious that both LR value and Wald value reject the null hypothesis. Therefore, SDM model is suitable to study the effect of industrial agglomeration on income inequality from a spatial perspective.

To choose between a fixed effects model and random effects model, many scholars use the Hausman test to evaluate whether there is a systematic difference between the coefficients by FE and RE [55]. Based on the Hausman test, this paper chooses the fixed effects model instead of random effects model. In addition, the correlation coefficients between independent variables and control variables are low, which shows they have no strong correlations and there is

**Table 4. Spatial econometric model testing.**

| Model | LM_Error | RLM_Error | LM_Lag | RLM_Lag | LR_SAR | LR_SEM | Wald_SAR | Wald_SEM |
|---|---|---|---|---|---|---|---|---|
| Spatial model | 51.348*** | 61.492*** | 0.155 | 10.3*** | 75.11*** | 68.18*** | 81.53*** | 73.22*** |

Note:

*** $p < 0.01$,

** $p < 0.05$,

* $p < 0.1$.

no multicollinearity. Thus, our proposed models are expressed as follows.

$$
\begin{aligned}
inequ_{it} = {} & \rho \sum\nolimits_{j=1}^{n} W_{ij} inequ_{it} + al_{it} + \beta_1 magg_{it} + \beta_2 magg_{it}^2 + \\
& \theta \sum\nolimits_{j=1}^{n} W_{ij} magg_{it} + \delta_1 Inpergdp_{it} + \delta_2 Inpergdp_{it}^2 + \delta_3 umemp_{it} + \\
& \delta_4 InGDR_{it} + \delta_5 human_{it} + \delta_6 gov_{it} + \delta_7 Insize_{it} + \\
& \varphi_1 \sum\nolimits_{j=1}^{n} W_{ij} Inpergdp_{it} + \varphi_2 \sum\nolimits_{j=1}^{n} W_{ij} gov_{it} + \varepsilon_{it}
\end{aligned}
\tag{8}
$$

## Results and discussion

### Empirical results

Based on the previous section, this paper uses fixed effects of the spatial Durbin model (SDM) to study the impact of manufacturing industrial agglomeration on regional income inequality. The results are shown in Table 5. Among them, (I), (II) and (III) use spatial Durbin model (SDM), and (IV) adopts ordinary least square (OLS). In regression (I), independent variables include industrial agglomeration (*magg*) and its quadratic term (magg$^2$) to test whether income inequality and industrial agglomeration are the non-linear relationship. Additionally, it adds a control variable, regional economic development level (*lnpergdp*). $R^2$ value in regression (I) is only 0.2788, which explains that the goodness of fit is relatively low. Regression (II) adds the quadratic term of regional economic development level (*lnpergdp*$^2$) to investigate whether "Kuznets curve" exists. Its goodness of fit is not the best. Therefore, in regression (III), we add more control variables, including unemployment rate (*unemp*), gross dependency ratio (*lnGDR*), human capital (*human*), government expenditure scale (*gov*) and population size (*lnsize*). It can be seen that $R^2$ value in regression (III) is higher than those in regression (I), (II) and (IV), indicating that the goodness of fit in (III) is the best and that SDM model used in this paper has strong explanatory power. In Table 5, the spatial coefficients $\rho$ pass the test at a significance level of 1% and they are significantly positive, which means that the spatial Durbin model estimation is effective. What's more, income inequality in neighboring provinces has a positive spillover impact on the province under study. Specifically, in regression (III), if the value of income inequality in neighboring provinces increases by 0.1, that in the province under study will increase 0.0351.

The values of the coefficients of the independent variables *magg* and *magg*$^2$ in Table 5 are significant in regression (I), (II) and (III), especially at the 1% level in (III). Although the independent variables in regression (IV) are not significant, the signs of *magg* and *magg*$^2$ are the same as (I), (II) and (III). Therefore, regarding the effect of manufacturing industrial agglomeration on income inequality, the results show that industrial agglomeration and income inequality present an inverted "U-shape" relationship, which proves that they are the non-linear change. As the degree of manufacturing industrial agglomeration increases, income inequality will rise, and when it reaches a certain value, income inequality will drop as the degree of industrial agglomeration increases. According to the calculation of the data in regression (III), when industrial agglomeration is estimated to be approximately 1.1638, the effect of industrial agglomeration on income inequality reaches its maximum. Holding control variables fixed, if the degree of industrial agglomeration increases from 0 to 1, the predicted rise in income inequality is 0.189 based on regression (III). However, if the degree of industrial agglomeration increases from 1 to 2, the predicted rise in income inequality is just 0.0266. Additionally, regression (I), (II) and (III) add the spatial spillover effect of manufacturing

**Table 5. The impacts of manufacturing industrial agglomeration on income inequality.**

| *Inequ* | (I) | (II) | (III) | (IV) |
|---|---|---|---|---|
| *Magg* | 0.148* | 0.267*** | 0.189*** | 0.366* |
|  | (1.90) | (3.80) | (2.89) | (1.81) |
| magg$^2$ | -0.0599* | -0.121*** | -0.0812*** | -0.133 |
|  | (-1.81) | (-4.01) | (-2.91) | (-1.18) |
| *Lnpergdp* | 0.347*** | 1.643*** | 1.211*** | 0.969** |
|  | (15.15) | (15.27) | (8.07) | (2.09) |
| lnpergdp$^2$ |  | -0.0672*** | -0.0476*** | -0.0465* |
|  |  | (-12.24) | (-6.26) | (-1.92) |
| *Unemp* |  |  | -0.0247*** | -0.0241* |
|  |  |  | (-3.81) | (-2.03) |
| *lnGDR* |  |  | 0.0805*** | 0.0618 |
|  |  |  | (3.58) | (0.90) |
| *Human* |  |  | -0.00930 | -0.00849 |
|  |  |  | (-1.57) | (-0.70) |
| *Gov* |  |  | -0.170*** | -0.178 |
|  |  |  | (-3.00) | (-1.56) |
| *Lnsize* |  |  | -0.370*** | -0.659*** |
|  |  |  | (-6.85) | (-3.29) |
| *Cons* |  |  |  | 1.123 |
|  |  |  |  | (0.31) |
| *P* | 0.549*** | 0.446*** | 0.351*** |  |
|  | (14.25) | (11.44) | (8.94) |  |
| *W*magg* | 0.266*** | 0.203*** | 0.172*** |  |
|  | (6.35) | (5.30) | (4.58) |  |
| *W*lnpergdp* | -0.343*** | -0.315*** | -0.254*** |  |
|  | (-14.59) | (-14.86) | (-11.60) |  |
| *W*gov* |  |  | 0.222*** |  |
|  |  |  | (3.01) |  |
| $R^2$ | 0.2788 | 0.4568 | 0.5868 | 0.5188 |

Note: t statistics in parentheses.

*** $p < 0.01$,

** $p < 0.05$,

* $p < 0.1$.

industrial agglomeration. It is obvious that the spatial spillover effect of industrial agglomeration is significantly positive at the 1% level. This means that industrial agglomeration in the neighboring provinces will promote income inequality in the province under study.

In Table 5, the signs of all control variables, either SDM model or OLS model, in their effects on income inequality are the same. Based on regression (III), the impact of regional economic development level on income inequality is significant at the 1% level and it presents an inverted "U-shape" change, which is consistent with "Kuznets curve". Unemployment rate, government expenditure scale and population size have significantly negative effects on income inequality at the 1% level, indicating that these factors can dampen the growth of income inequality. The effect of gross dependency ratio on income inequality is significantly positive at the 1% level, which is in line with the actual situation. The impact of human capital on income inequality in China is not significant according to Table 5. In addition, regression

(III) adds the spatial spillover effects of two control variables, including regional economic development level and government expenditure scale. The result shows that the spatial spillover effect of regional economic development level is significantly negative at the 1% level, which suggests that economic development level in the surrounding provinces contributes to reduce income inequality in the province under study. The spatial spillover effect of government expenditure scale is positive at a significance level of 1%, indicating that government expenditure scale in the surrounding provinces will increase income inequality in the province under analysis.

## Robustness test

This paper will conduct a robustness test from two aspects, replacing independent variables and the spatial weight matrix. First, the robustness test is performed by selecting other indicators as explanatory variables and the independent variable manufacturing industrial agglomeration (*magg*) is replaced with manufacturing employment density (*mden*) which is the ratio of manufacturing employment to the area. Furthermore, based on the empirical evidence of Zhang et al. [56], including the one-period lagged independent and control variables can alleviate the potential endogeneity problem. Thus, the robustness is also tested by adopting the one-period lagged manufacturing industrial agglomeration ($magg_{t-1}$) instead of manufacturing industrial agglomeration. In addition, some scholars believe that different spatial weight matrices can be constructed to verify whether the spatial model design is reasonable [57]. Thus, this paper introduces the economic distance weight matrix to test its robustness. The economic distance weight matrix: the main diagonal elements are 0. (i, j) of the non-main diagonal is $W_{ij} = \frac{1}{\bar{Y}_i - \bar{Y}_j}$ (i≠j), $\bar{Y}_i$ is the average real GDP per capita of region i in the sample from 2003 to 2017. $\bar{Y}_j$ is the average real GDP per capita of region j in the sample. The results are shown in Table 6 Regression (III) is the estimation of the above SDM model (III) in Table 5. Regression (V) and (VI) are the estimations after replacing the independent variables by the one-period lagged manufacturing industrial agglomeration and manufacturing employment density respectively. Regression (VII) is the estimation after changing the spatial weight matrix using the economic distance weight matrix. *W\*magg*, $W*magg_{t-1}$, *W\*lnmden*, *W\*lnpergdp* and *W\*gov* are the spatial coefficients. The estimation results of regression (III), (V), (VI) and (VII) show that all of the spatial coefficients $\rho$ pass the test at the 1% significance level, which indicates that the four spatial models are effective. Regression (V) and (VI) show that the coefficients of the one-period lagged manufacturing industrial agglomeration and manufacturing employment density are significant, proving the results are still robust. The signs of the coefficients of *magg* and *magg*$^2$ in regression (III) and (VII) are the same and they are all significant at the 1% level, indicating again that manufacturing industrial agglomeration and income inequality are the non-linear relationship.

## Discussion

This study is aimed at investigating the impact of manufacturing industrial agglomeration on regional income inequality in China. Firstly, we adopt a relative value, the proportion of the per capita income of each province to the national per capita income, as the dependent variable. When the dependent variable is closer to 1, it shows the more equal regional income, which is also our goal. Based on Moran's *I* of income inequality, this paper finds that income inequality in different provinces of China is not randomly distributed, but has a significant positive spatial correlation. This result is consistent with Peng and Yuan [30] and Wang and Liu [28] who examined the spatial autocorrelation of urban and rural income inequality in

**Table 6. Robustness test of the spatial Durbin model.**

| *Inequ* | (III) | (V) | (VI) | (VII) |
|---|---|---|---|---|
| *Magg* | 0.189*** | | | 0.163*** |
| | (2.89) | | | (2.65) |
| $magg^2$ | -0.0812*** | | | -0.0764*** |
| | (-2.91) | | | (-2.91) |
| $magg_{t-1}$ | | 0.169** | | |
| | | (2.55) | | |
| $magg^2_{t-1}$ | | -0.0752*** | | |
| | | (-2.67) | | |
| *lnmden* | | | 0.0271* | |
| | | | (1.92) | |
| $lnmden^2$ | | | -0.0173*** | |
| | | | (-7.25) | |
| *lnpergdp* | 1.211*** | 1.405*** | 0.933*** | 0.938*** |
| | (8.07) | (8.40) | (6.24) | (6.26) |
| $lnpergdp^2$ | -0.0476*** | -0.0568*** | -0.0347*** | -0.0354*** |
| | (-6.26) | (-6.75) | (-4.63) | (-4.64) |
| *Unemp* | -0.0247*** | -0.0242*** | -0.0282*** | -0.0181*** |
| | (-3.81) | (-3.61) | (-4.48) | (-2.91) |
| *lnGDR* | 0.0805*** | 0.0810*** | 0.0502** | 0.0836*** |
| | (3.58) | (3.52) | (2.10) | (3.82) |
| *human* | -0.00930 | -0.00768 | -0.00953 | -0.00291 |
| | (-1.57) | (-1.27) | (-1.63) | (-0.51) |
| *Gov* | -0.170*** | -0.157*** | -0.195*** | -0.0234 |
| | (-3.00) | (-2.77) | (-3.53) | (-0.46) |
| *Lnsize* | -0.370*** | -0.350*** | -0.472*** | -0.438*** |
| | (-6.85) | (-6.32) | (-8.74) | (-8.63) |
| *P* | 0.351*** | 0.365*** | 0.299*** | 0.308*** |
| | (8.94) | (8.97) | (7.48) | (4.93) |
| *W\*magg* | 0.172*** | | | 0.500*** |
| | (4.58) | | | (7.47) |
| $W*magg_{t-1}$ | | 0.168*** | | |
| | | (4.51) | | |
| *W\*lnmden* | | | 0.0572*** | |
| | | | (3.64) | |
| *W\*lnpergdp* | -0.254*** | -0.261*** | -0.233*** | -0.206*** |
| | (-11.60) | (-11.22) | (-10.20) | (-10.63) |
| *W\*gov* | 0.222*** | 0.194*** | 0.140* | |
| | (3.01) | (2.59) | (1.91) | |
| $R^2$ | 0.5868 | 0.5828 | 0.6308 | 0.6536 |

Note: t statistics in parentheses.

*** $p < 0.01$,

** $p < 0.05$,

* $p < 0.1$.

China. Most previous studies on income inequality, such as Bengtsson and Waldenström [58] and Heimberger [59], had paid little attention to its spatiality. However, in our study, both the spatial coefficient of income inequality and its Moran's $I$ are positive. The result indicates that income inequality in a province will be affected by income inequality of neighboring provinces.

Secondly, in our study, the impact of manufacturing industrial agglomeration on regional income inequality shows an inverted U-shape relationship which is similar to the result of Wang and Zhou [24]. In the initial stage, there is a positive correlation between manufacturing industrial agglomeration and income inequality. As industrial agglomeration increases, regional income inequality also rises. This is mainly because the agglomeration effect of manufacturing industrial agglomeration in the initial stage will be beneficial to the increase of per capita income of some regions, exceeding the national per capita income, while other regions are not so lucky, and the per capita income of these regions will be lower than the national per capita income, which results in an increase in income inequality in China as industrial agglomeration increases. However, when industrial agglomeration reaches a certain value, industrial agglomeration will contribute to reduce regional income inequality, because the crowding effect will dominate instead of the agglomeration effect. When some regions develop to a certain extent, limited resources will promote industrial agglomeration to spread to other regions, increase per capita income of other regions, and thus effectively reduce regional income inequality. The result is not consistent with most of literatures, such as Peng and Yuan [30] and Tao [25]. They ignored the nonlinear relationship between industrial agglomeration and income inequality. Additionally, manufacturing industrial agglomeration in the neighboring regions will increase income inequality in the region. This is because unbalanced industrial agglomeration in and around the region will contribute to income inequality in the region.

Thirdly, based on the results of the control variables, unemployment rate, government expenditure scale and population size have significantly negative effects on income inequality, explaining that these three factors can help reduce income inequality. The result of government expenditure scale is in line with that of Dunford and Perrons [36], Charron [37] and Iammarino et al. [38]. However, the effects of unemployment rate and population size on income inequality are not consistent with most of studies. This is because our study uses the proportion of the per capita income of each province to the national per capita income as income inequality. When unemployment rate and population size increase, it will dilute per capita income, thereby reducing income inequality in a country. The result of regional economic development level on income inequality shows an inverted "U-shape" change, which is consistent with Kuznets [35] and Soava et al. [60] but not consistent with Kavya and Shijin [61] who thought that only high-income countries support the existence of a Kuznets curve while both middle and low-income countries support U shaped pattern between economic development and income inequality. The effect of gross dependency ratio on income inequality is positive, which is in line with Breau [22]. He believed that the greater shares of the elderly and young will put pressure on the work force, resulting in the increase of income inequality. The spatial spillover effect of regional economic development level and government expenditure scale are in line with the actual situation.

## Conclusions

Industrial agglomeration has accelerated economic growth, but it may also widen regional income inequality to a certain extent. This paper explores the effect of manufacturing industrial agglomeration on regional income inequality from a spatial perspective and measures

their spatial relationship. According to the literatures about the relationship of industrial agglomeration and income inequality, using data from 31 provinces in China from 2003 to 2020, this study verifies and discusses the impact of manufacturing industrial agglomeration on income inequality. The following suggestions and limitations can be drawn.

When we attempt to cut down income inequality in this study, the values of income inequality in 31 provinces should be close to 1. How to make income inequality approach 1? First, based on Table 1, the values of income inequality more than 1 are concentrated in coastal provinces, while those less than 1 focus on inland provinces. Additionally, the spatial distribution of industrial agglomeration in 2020 is similar to that of income inequality in 2020, which shows that manufacturing industrial agglomeration focuses on coastal provinces. Thus, to reduce regional income inequality, the Chinese government should transfer some manufacturing industrial agglomerations in coastal provinces to inland provinces, which can cultivate and develop manufacturing industries in inland provinces and improve their per capita income. Second, from the previous sections, the relationship of industrial agglomeration and income inequality presents an inverted U-shape relationship. Therefore, the coastal provinces should exert the diffusion effect of their manufacturing industrial agglomeration, drive the development of the surrounding provinces, and better improve the per capita income of the surrounding provinces. These provinces had better improve the quality and efficiency of their manufacturing industries. They should strive to develop high-end manufacturing so as to drive the qualitative leap of the entire Chinese manufacturing industry. Third, all provinces had better try their best to continue to improve regional economic development level, increase government expenditure scale and lower gross dependency ratio to effectively reduce income inequality. At the same time, the Chinese government should pay attention to balancing government expenditure scale in various regions in order to avoid the spatial spillover effect of government expenditure scale leading to an increase in income inequality.

The findings of this paper could be used to investigate the impacts of manufacturing industrial agglomeration on income inequality in other countries which suffer from regional income inequality. However, this paper exists its limitations. First, we use the proportion of the per capita income of each province to the national per capita income as the dependent variable because it is a good proxy for regional income inequality. We do not use the commonly used proxies for income inequality, such as the Gini coefficient and the Theil index and so on. When studying other determinants of income inequality, such as unemployment rate, we can consider the Gini coefficient as income inequality in the future studies. Second, we do not use other methods to calculate the degree of manufacturing industrial agglomeration, such as the spatial Gini coefficient, which could have different influences on regional income inequality. Hence, future studies on the degree of manufacturing industrial agglomeration may want to consider other methods.

## Acknowledgments

The authors would like to thank Gareth and Sarocha for their language assistance and the anonymous reviewers for their constructive comments.

## Author Contributions

**Writing – original draft:** Suhua Zhang.

**Writing – review & editing:** Yasmin Bani, Aslam Izah Selamat, Judhiana Abdul Ghani.

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
