## [Decision Letter · Decision Letter 0]

2 Jan 2023

PONE-D-22-32289Exploring the effect of industrial agglomeration on income inequality in ChinaPLOS ONE

Dear Dr. Zhang,

Thank you for submitting your manuscript to PLOS ONE. After careful consideration, we feel that it has merit but does not fully meet PLOS ONE’s publication criteria as it currently stands. Therefore, we invite you to submit a revised version of the manuscript that addresses the points raised during the review process.

We look forward to receiving your revised manuscript.

Kind regards,

Bing Xue, Ph.D.

Academic Editor

PLOS ONE

Journal Requirements:

5. We note that Figure 1 in your submission contain map images which may be copyrighted. All PLOS content is published under the Creative Commons Attribution License (CC BY 4.0), which means that the manuscript, images, and Supporting Information files will be freely available online, and any third party is permitted to access, download, copy, distribute, and use these materials in any way, even commercially, with proper attribution. For these reasons, we cannot publish previously copyrighted maps or satellite images created using proprietary data, such as Google software (Google Maps, Street View, and Earth). For more information, see our copyright guidelines: http://journals.plos.org/plosone/s/licenses-and-copyright.

(1) You may seek permission from the original copyright holder of Figure 1 to publish the content specifically under the CC BY 4.0 license.  

Reviewers' comments:

Reviewer's Responses to Questions

**Comments to the Author**

1. Is the manuscript technically sound, and do the data support the conclusions?

Reviewer #1: Yes

Reviewer #2: Yes

2. Has the statistical analysis been performed appropriately and rigorously? 

Reviewer #1: I Don't Know

Reviewer #2: Yes

3. Have the authors made all data underlying the findings in their manuscript fully available?

Reviewer #1: Yes

Reviewer #2: Yes

4. Is the manuscript presented in an intelligible fashion and written in standard English?

Reviewer #1: Yes

Reviewer #2: Yes

5. Review Comments to the Author

Reviewer #1: Income inequality is one of the major social issues in China at this stage, and industrial agglomeration helps to promote the mobility of technology and people. This manuscript explores the nonlinear relationship and spatial effects of industrial agglomeration from the perspective of its association with income inequality.The manuscript has content compliance, system integrityand reasonable result. However, there are still some insufficiencies:

1.Abstract: The abstract section should contain the research background, motivation, questions, methods, conclusions and insights, etc. The elaboration of the research gap can be briefly elaborated or deleted.

2.Introduction: The introduction section, as the beginning of the article, answers the question of "why research". It is enough to explain the background, the history and status of previous research in the relevant field and the research gap, the value and the purpose of the research, the author uses five paragraphs to elaborate, which will lead to too much content and illogicality in this section, please delete unnecessary content. In addition, the introduction section lacks a description of the research objectives.Then, please add the corresponding references.e.g.line80-line82.

3.Literature review: In the literature review section, the author divided into three parts, the definition and measurement of industrial agglomeration, the relationship between industrial agglomeration and income inequality, and other factors that affect income inequality. The content is abundant, but it does not summarize and condense the literature combing, the literature is seriously piled up, and the classification of each scholar's view is not sufficiently summarized.Then, please add the corresponding references.e.g.line146.

4.Data and methodology : In the variable construction section, where the author compares and contrasts measures of income inequality, please insert appropriate citations to enhance the power of persuasion. In the section on control variables, please give appropriate citations to justify and demonstrate their usefulness. Also, in the “Other determinants of income inequality section” of the literature review, the factors mentioned that affect income inequality also include “economic growth, human capital, transportation, economic structure, government intervention, and unemployment”. Do they have any influence on this research and are they all included in the consideration of the control variables. This leads me to question the scientific validity and reasonableness of the third part of the literature review in the manuscript. The format of equation (2) and equation (3) is not uniform, please revise. The p-value of the indicator used to express the significance of the variables should be italicized, pleace check and correct it. In addition, I suggest that the spatial correlation analysis be placed in the results and discussion section.

5.Results and discussion: First, discuss results in detail and mention the innovative outcome. Further results should be backed with appropriate literature. Results should be further elaborated in detail. Usually, relevant tests need to be performed on the data before conducting empirical analysis. Considering that this manuscript uses data from 31 provinces in China, here is a question: When the authors processed the data, were the data for Tibet, Hong Kong, Macau and Taiwan complete? If not, how were you processed? If non-equilibrium panel data are used, do the corresponding tests of the data meet the requirements? Please show the results of the tests in the manuscript to prove the scientific validity of the manuscript. Moreover, Table 3 demonstrates a low R2, which confirms the possible inappropriateness of the author's choice of control variables.Also, please fix the formatting problems in Table 3.

6.Conclusions: Please differentiate with the discussion sectionCondense the core conclusions of this manuscript and delete some contents that belong to the discussion section.

7.References: Too many references cited, please make appropriate deletions.

8.Avoid grammatical and typo errors and revise the manuscripts for corresponding concerns.

9.Corresponding author information needs to be confirmed. (title page )

Reviewer #2: 1. Manuscript needs to follow the journal guideline and template.

2. Use initial capitals for each key word and separate them with a comma.

3. Try to start the abstract by interdicting why are you envisioned to do this research, the abstract is very long try to write it in a precise and comprehensive way.

4. Headings mentioned in the literature section are not appropriate, try to rewrite. E.g, “Defining and measuring industrial agglomeration” this heading can be written as “Measurement of industrial agglomeration”. Instead of “The link between industrial agglomeration and income inequality” you can write “Linkages between industrial agglomeration and income inequality”. “Other determinants of income inequality” “income inequality and its determinants “.

5. Abbreviations should be written in full form at the first place. E.g GE coefficient is not clear.

6. Instead of writing “Table 1. Data”, write “Table 1. Variable descriptions”.

7. An explanation about the method used to analyze the data in the study is missing.

6. PLOS authors have the option to publish the peer review history of their article (what does this mean?). If published, this will include your full peer review and any attached files.

Reviewer #1: No

Reviewer #2: No

---

## [Author Response · Author response to Decision Letter 0]

15 Feb 2023

February 8, 2023

PLOS ONE

Dear Editors and Reviewers:

Thank you for your letter and for the reviewers’ comments concerning our manuscript (PONE-D-22-32289). Those comments are all valuable and very helpful for revising and improving our paper, as well as the important guiding significance to our researches. We have studied comments carefully and have made correction which we hope meet with approval. Revised portion are marked up using the “Track Changes” function in the paper. The main corrections in the paper and the responses to the reviewer’s comments are as following:

Response to Reviewer 1 Comments:

Point 1: Abstract: The abstract section should contain the research background, motivation, questions, methods, conclusions and insights, etc. The elaboration of the research gap can be briefly elaborated or deleted.

Response 1: Based on the reviewer’s comments, we have deleted “A number of studies show that if income inequality in a country is too large, it will not be conducive to the development and growth of the country, and will result in great social and political instability” and “Industrial agglomeration has accelerated economic growth, but it may also widen regional income inequality” which are not very important when we elaborate the gap in the abstract. Thanks for your valuable comments.

Point 2: Introduction: The introduction section, as the beginning of the article, answers the question of "why research". It is enough to explain the background, the history and status of previous research in the relevant field and the research gap, the value and the purpose of the research, the author uses five paragraphs to elaborate, which will lead to too much content and illogicality in this section, please delete unnecessary content. In addition, the introduction section lacks a description of the research objectives. Then, please add the corresponding references.e.g.line80-line82.

Response 2: We gratefully appreciate for your comment. We have deleted some unnecessary content. Regarding the research objective, it can be reflected in line75-line76. Additionally, several contributions in line93-line114 also explain the goals of this paper. The corresponding references in contributions have been elaborated in the following sections. Thanks for your valuable comments.

Point 3: Literature review: In the literature review section, the author divided into three parts, the definition and measurement of industrial agglomeration, the relationship between industrial agglomeration and income inequality, and other factors that affect income inequality. The content is abundant, but it does not summarize and condense the literature combing, the literature is seriously piled up, and the classification of each scholar's view is not sufficiently summarized. Then, please add the corresponding references.e.g.line146.

Response 3: We appreciate for your valuable comments. In the literature section, the beginning of each paragraph summarizes the content and reflects the classification of scholars’ view. According to the reviewer’s suggestions, we have rewritten the content in line190-line213. The literature by Maure and Sedilbt (1999) is not important in the paper. Therefore, we have deleted this literature based on the reviewer’s comments. Thanks for your valuable comments.

Point 4: Data and methodology : In the variable construction section, where the author compares and contrasts measures of income inequality, please insert appropriate citations to enhance the power of persuasion. In the section on control variables, please give appropriate citations to justify and demonstrate their usefulness. Also, in the “Other determinants of income inequality section” of the literature review, the factors mentioned that affect income inequality also include “economic growth, human capital, transportation, economic structure, government intervention, and unemployment”. Do they have any influence on this research and are they all included in the consideration of the control variables. This leads me to question the scientific validity and reasonableness of the third part of the literature review in the manuscript. The format of equation (2) and equation (3) is not uniform, please revise. The p-value of the indicator used to express the significance of the variables should be italicized, please check and correct it. In addition, I suggest that the spatial correlation analysis be placed in the results and discussion section.

Response 4: It is really true as Reviewer suggested. We have added citations to enhance the power of persuasion in line298-line301. Appropriate citations on control variables are written in the literature’s third part “Income inequality and its determinants” and the model specification’s line470-line473. Most of factors including “economic growth, human capital, transportation, economic structure, government intervention, and unemployment and so on” are considered in the control variables. For example, economic growth elaborated in the literature section is also expressed by regional economic development level in our model which is a very important control variable. However, some factors such as transportation are not significant in our empirical study in China although maybe significant in other countries. The choice of control variables in this study is seriously checked and based on China’s situation and some important literature. We appreciate for your valuable suggestions about the place of the spatial correlation analysis. In this paper, the empirical model, the spatial Durbin model is put forward after proving that dependent variable or independent variable is correlated spatially. Thus, we made the spatial correlation analysis be placed before the model specification section. Thanks for your valuable comments.

Point 5: Results and discussion: First, discuss results in detail and mention the innovative outcome. Further results should be backed with appropriate literature. Results should be further elaborated in detail. Usually, relevant tests need to be performed on the data before conducting empirical analysis. Considering that this manuscript uses data from 31 provinces in China, here is a question: When the authors processed the data, were the data for Tibet, Hong Kong, Macau and Taiwan complete? If not, how were you processed? If non-equilibrium panel data are used, do the corresponding tests of the data meet the requirements? Please show the results of the tests in the manuscript to prove the scientific validity of the manuscript. Moreover, Table 3 demonstrates a low R2, which confirms the possible inappropriateness of the author's choice of control variables. Also, please fix the formatting problems in Table 3.

Response 5: We appreciate for your valuable comments. There are some appropriate literature to support the further results in the discussion section. The data are for Tibet but not for Hong Kong, Macau and Taiwan. Because Hong Kong, Macau and Taiwan have different systems with China mainland. Additionally, we used equilibrium panel data in our study which include 31 provinces from 2003 to 2020. Therefore, the corresponding tests of the data meet the requirements. Regarding R2, when we chose the appropriate control variables, we have run the regressions from (I) to (III) and the R2 is improved greatly from 0.2788 to 0.5868, which proves the possible appropriateness of control variables. We have fixed the formatting problems in all tables. Thanks for your valuable comments.

Point 6: Please differentiate with the discussion section, condense the core conclusions of this manuscript and delete some contents that belong to the discussion section.

Response 6: We have deleted some not important contents in the conclusions section. Thanks for your valuable comments.

Point 7: References: Too many references cited, please make appropriate deletions.

Response 7: We have deleted some not important references based on the reviewer’s comments. Thanks for your valuable comments.

Point 8: Avoid grammatical and typo errors and revise the manuscripts for corresponding concerns.

Response 8: We have checked and revised the entire paper. Thanks for your valuable comments.

Point 9: Corresponding author information needs to be confirmed. (title page )

Response 9: The title page has been included into the beginning of our manuscript. There are two corresponding authors which have been added in the title page. Thanks for your valuable comments.

Response to Reviewer 2 Comments:

Point 1: Manuscript needs to follow the journal guideline and template.

Response 1: Based on the journal guideline and template, we have checked and revised the entire paper. Thanks for your valuable comments.

Point 2: Use initial capitals for each key word and separate them with a comma.

Response 2: We have used initial capitals for each key word and separate them with a comma based on the reviewer’s comments. Thanks for your valuable comments.

Point 3: Try to start the abstract by interdicting why are you envisioned to do this research, the abstract is very long try to write it in a precise and comprehensive way.

Response 3: We have deleted “A number of studies show that if income inequality in a country is too large, it will not be conducive to the development and growth of the country, and will result in great social and political instability” and “Industrial agglomeration has accelerated economic growth, but it may also widen regional income inequality” to make the abstract shorter according to the reviewer’s comments. Thanks for your valuable comments.

Point 4: Headings mentioned in the literature section are not appropriate, try to rewrite. E.g, “Defining and measuring industrial agglomeration” this heading can be written as “Measurement of industrial agglomeration”. Instead of “The link between industrial agglomeration and income inequality” you can write “Linkages between industrial agglomeration and income inequality”. “Other determinants of income inequality” “income inequality and its determinants”.

Response 4: We appreciate for your valuable suggestions. Thus, we have rewritten the headings in the literature section based on your comments. Thanks for your valuable comments.

Point 5: Abbreviations should be written in full form at the first place. E.g GE coefficient is not clear.

Response 5: Considering the Reviewer’s suggestion, we have added the full form of abbreviations at the first place in line162, line170-line171, line290-line291 and line327. Thanks for your valuable comments.

Point 6: Instead of writing “Table 1. Data”, write “Table 1. Variable descriptions”.

Response 6: Based on the reviewer’s suggestion, we have rewritten the title of this table. Thanks for your valuable comments.

Point 7: An explanation about the method used to analyze the data in the study is missing.

Response 7: An explanation about the method used to analyze the data is written in the section of “Data and Methodology” and concluded in line476-line486. Thanks for your valuable comments.

Other changes:

1. There are two corresponding authors, Suhua Zhang and Yasmin Bani. We have revised this information in the title page. Additionally, the title page has been included into the beginning of our manuscript based on your requirements.

2. The data are publicly accessible at (doi:10.5061/dryad.z08kprrht).

3. Based on the editor’s suggestions, we have changed the Figure which contains map images to Table 1 in line318-line357.

4. We have deleted some not important references in this paper according to your comments.

5. We have checked and revised the formatting problems based on the PLOS ONE style templates.

We appreciate for editors and reviewers’ warm work earnestly, and hope that the correction will meet with approval. Once again, thank you very much for your comments and suggestions.

Sincerely,

Suhua Zhang

zhangsuhua1023@gmail.com

---

## [Decision Letter · Decision Letter 1]

16 May 2023

PONE-D-22-32289R1Exploring the effect of industrial agglomeration on income inequality in ChinaPLOS ONE

Dear Dr. Zhang,

Thank you for submitting your manuscript to PLOS ONE. After careful consideration, we feel that it has merit but does not fully meet PLOS ONE’s publication criteria as it currently stands. Therefore, we invite you to submit a revised version of the manuscript that addresses the points raised during the review process.

We look forward to receiving your revised manuscript.

Kind regards,

Fuyou Guo, (Ph.D.

Academic Editor

PLOS ONE

Journal Requirements:

Additional Editor Comments:

Reviewer Comments:

Point 1: confusion of important concepts. For example, “Industrial agglomeration” and “Industrial cluster” are not the same concept. The author must carefully read some important literature about “industrial agglomeration”, and enhance the ability of concept discrimination. What’s more, some crucial statements lack preciseness. For example, “few have studied the impacts of industrial agglomeration on income inequality, and even fewer have studied the spatial correlation of income inequality”.

Point 2: Literature review: the part of “Measurement of industrial agglomeration” needs to be simplified. This part focuses on the main measurement methods of industrial agglomeration (brief introduction), “why does this paper choose ‘location quotient’ to measure industrial agglomeration”, and other important content. The part of “Linkages between industrial agglomeration and income inequality”: It is suggested that the logical relationship and mechanism between them should be effectively analyzed, rather than the stacking statement of the existing literature. The part of “Income inequality and its determinants”: why write this part? Some variables are not effectively reflected in the empirical analysis of this paper. So, it is not recommended to state them as a separate part. What’s more, the Literature review should focus on "income inequality ", "industrial agglomeration" and "the relationship between them".

Point3: Results and discussion: why was SPDM chosen as the empirical model of this paper? There are many models for testing nonlinear relationship. “Spatial effect of industrial agglomeration on income inequality” should be supplemented, especially, in the part of literature review. Lack of necessary testing analysis (e.g., Likelihood ratio test, Wald test, and etc.). The robustness test is unreasonable. Why is “service industry agglomeration” chosen to replace the independent variable? Does industry heterogeneity need further consideration? Why not choose other variables (e.g., manufacturing employment density) to substitute for the independent variable? You can also select some variables to substitute for the explained variable. The regression model of industrial agglomeration affecting income inequality will inevitably be troubled by the endogeneity problem to a certain extent. So, Endogeneity test may be warranted. Standardization also needs to be strengthened.

Piont4: The paper is not innovative enough, and the research conclusions are not novel enough.

Reviewers' comments:

Reviewer's Responses to Questions

**Comments to the Author**

1. If the authors have adequately addressed your comments raised in a previous round of review and you feel that this manuscript is now acceptable for publication, you may indicate that here to bypass the “Comments to the Author” section, enter your conflict of interest statement in the “Confidential to Editor” section, and submit your "Accept" recommendation.

Reviewer #2: All comments have been addressed

Reviewer #3: (No Response)

2. Is the manuscript technically sound, and do the data support the conclusions?

Reviewer #2: Yes

Reviewer #3: (No Response)

3. Has the statistical analysis been performed appropriately and rigorously? 

Reviewer #2: Yes

Reviewer #3: (No Response)

4. Have the authors made all data underlying the findings in their manuscript fully available?

Reviewer #2: Yes

Reviewer #3: (No Response)

5. Is the manuscript presented in an intelligible fashion and written in standard English?

Reviewer #2: Yes

Reviewer #3: (No Response)

6. Review Comments to the Author

Reviewer #2: Authors have improved the manuscript and the current version is good to be published in Plos One.

Good luck.

Reviewer #3: Point 1: confusion of important concepts. For example, “Industrial agglomeration” and “Industrial cluster” are not the same concept. The author must carefully read some important literature about “industrial agglomeration”, and enhance the ability of concept discrimination. What’s more, some crucial statements lack preciseness. For example, “few have studied the impacts of industrial agglomeration on income inequality, and even fewer have studied the spatial correlation of income inequality”.

Point 2: Literature review: the part of “Measurement of industrial agglomeration” needs to be simplified. This part focuses on the main measurement methods of industrial agglomeration (brief introduction), “why does this paper choose ‘location quotient’ to measure industrial agglomeration”, and other important content. The part of “Linkages between industrial agglomeration and income inequality”: It is suggested that the logical relationship and mechanism between them should be effectively analyzed, rather than the stacking statement of the existing literature. The part of “Income inequality and its determinants”: why write this part? Some variables are not effectively reflected in the empirical analysis of this paper. So, it is not recommended to state them as a separate part. What’s more, the Literature review should focus on "income inequality ", "industrial agglomeration" and "the relationship between them".

Point3: Results and discussion: why was SPDM chosen as the empirical model of this paper? There are many models for testing nonlinear relationship. “Spatial effect of industrial agglomeration on income inequality” should be supplemented, especially, in the part of literature review. Lack of necessary testing analysis (e.g., Likelihood ratio test, Wald test, and etc.). The robustness test is unreasonable. Why is “service industry agglomeration” chosen to replace the independent variable? Does industry heterogeneity need further consideration? Why not choose other variables (e.g., manufacturing employment density) to substitute for the independent variable? You can also select some variables to substitute for the explained variable. The regression model of industrial agglomeration affecting income inequality will inevitably be troubled by the endogeneity problem to a certain extent. So, Endogeneity test may be warranted. Standardization also needs to be strengthened.

Piont4: The paper is not innovative enough, and the research conclusions are not novel enough.

7. PLOS authors have the option to publish the peer review history of their article (what does this mean?). If published, this will include your full peer review and any attached files.

Reviewer #2: No

Reviewer #3: No

---

## [Author Response · Author response to Decision Letter 1]

29 May 2023

May 30, 2023

PLOS ONE

Dear Editors and Reviewers:

We would like to thank the editor for giving us a chance to revise the manuscript, and thank the reviewers’ comments concerning our manuscript (PONE-D-22-32289R1). Those comments are all valuable and very helpful for revising and improving our paper, as well as the important guiding significance to our researches. We have studied comments carefully and have made correction which we hope meet with your approval. Revised parts are marked up using the “Track Changes” function in the paper. The main corrections in the paper and the responses to the reviewer’s comments are as following:

Response to Reviewer #2 Comments:

Point: Authors have improved the manuscript and the current version is good to be published in Plos One. Good luck.

Reply: We gratefully appreciate for your comments and encouragement.

Response to Reviewer #3 Comments:

Point 1: Confusion of important concepts. For example, “Industrial agglomeration” and “Industrial cluster” are not the same concept. The author must carefully read some important literature about “industrial agglomeration”, and enhance the ability of concept discrimination. What’s more, some crucial statements lack preciseness. For example, “few have studied the impacts of industrial agglomeration on income inequality, and even fewer have studied the spatial correlation of income inequality”.

Reply 1: Thanks to the reminding of the reviewer. Because our expression about the correlation between industrial agglomeration and industrial cluster is not accurate. Based on your suggestion, we have revised the content from “These three names are much the same” to “These three names are closely related concepts, but they have subtle differences”. Additionally, we have rewritten the content on “few have studied the impacts of industrial agglomeration on income inequality, and even fewer have studied the spatial correlation of income inequality”. The current content is “few studies have been conducted on the impacts of industrial agglomeration on income inequality and their spatial correlation”.

Point 2: Literature review: the part of “Measurement of industrial agglomeration” needs to be simplified. This part focuses on the main measurement methods of industrial agglomeration (brief introduction), “why does this paper choose ‘location quotient’ to measure industrial agglomeration”, and other important content. The part of “Linkages between industrial agglomeration and income inequality”: It is suggested that the logical relationship and mechanism between them should be effectively analyzed, rather than the stacking statement of the existing literature. The part of “Income inequality and its determinants”: why write this part? Some variables are not effectively reflected in the empirical analysis of this paper. So, it is not recommended to state them as a separate part. What’s more, the Literature review should focus on "income inequality ", "industrial agglomeration" and "the relationship between them".

Reply 2: Thanks for your valuable comments. The part of “Measurement of industrial agglomeration” concentrates on explaining why this paper choose ‘location quotient’ and introducing this method. Thus, we have added three literatures on location quotient and adjusted the expression in this part. The new content is “Although adopted by some scholars, the above methods are not suitable for this study because of the aforementioned shortcomings. Therefore, this paper employs the commonly used location quotient (LQ) to calculate the degree of industrial agglomeration. LQ is a very meaningful indicator to measure the spatial distribution of elements in a certain region and reflects the degree of specialization of a certain industrial sector. The greater the value of LQ, the greater the specialization rate. Munnich et al. (1998) adopted LQ as the measurement standard. Peters (2004) used LQ as the measurement standard. And he measured economic specialization for an industry in Missouri by calculating LQ for output, employment, compensation and foreign exports in 2000. Jiang and Xu (2016) utilized location quotient (LQ) to measure the level of forestry industry agglomeration in Heilongjiang of China from the two perspectives of gross product and number of employees. Zhang et al. (2016) employed LQ to measure the degree of industrial agglomeration taking industrial industries in different regions of China as research objects”. The main measurement methods of industrial agglomeration are also important. Because we discussed the shortcomings of every methods after introducing these methods, for example, “the D-O index has very strict data requirements”. Finally, we decided not to delete the content on main methods. Regarding the part of “Linkages between industrial agglomeration and income inequality”, based on your suggestion, we have slightly adjusted the paragraphs to make the summaries clear. Additionally, for the part of “Income inequality and its determinants”, we have deleted the content on some variables not reflected in the empirical analysis. We also revised some variables’ names, such as “economic growth” to “regional economic development level” because both of them actually adopt the same index, to be consistent with the variables names in the empirical analysis of this paper. Based on your suggestion, the modified part of “Income inequality and its determinants” has been merged into the back of the part “Linkages between industrial agglomeration and income inequality”.

Point 3: Results and discussion: why was SPDM chosen as the empirical model of this paper? There are many models for testing nonlinear relationship. “Spatial effect of industrial agglomeration on income inequality” should be supplemented, especially, in the part of literature review. Lack of necessary testing analysis (e.g., Likelihood ratio test, Wald test, and etc.). The robustness test is unreasonable. Why is “service industry agglomeration” chosen to replace the independent variable? Does industry heterogeneity need further consideration? Why not choose other variables (e.g., manufacturing employment density) to substitute for the independent variable? You can also select some variables to substitute for the explained variable. The regression model of industrial agglomeration affecting income inequality will inevitably be troubled by the endogeneity problem to a certain extent. So, Endogeneity test may be warranted. Standardization also needs to be strengthened.

Reply 3: Thanks for your valuable comments. There are several reasons mentioned in the paper why this study chooses the spatial Durbin model. First, according to Moran’s I, the spatial autocorrelation of dependent and independent variables exists. Second, the industrial agglomeration not only affects income inequality in the province, but also income inequality of surrounding provinces. Income inequality between neighboring provinces also has spatial correlation and spatial spillover effects. Third, based on the suggestions of the reviewer, we have added testing analysis, such as Likelihood ratio test and Wald test, to verify why the spatial Durbin model is adopted. We present it in Table 4. (added manuscript: Additionally, a combination of LM_Error, RLM_Error (spatial error robustness test), LM_Lag and RLM_Lag (spatial lag robustness test) is adopted to further validate why this study uses the spatial Durbin model (SDM) rather than other common spatial models. The results are shown in Table 4. Based on the model without spatial effect (except LM_Lag), all the null hypotheses are rejected. Thus, SDM model is usually given priority because both the spatial autoregressive model (SAR) and the spatial errors model (SEM) can be accepted (Ellison et al., 2010). This paper also uses LR test and Wald test to prove whether SDM model can be degenerated into SAR model or SEM model. It is obvious that both LR value and Wald value reject the null hypothesis. Therefore, SDM model is suitable to study the effect of industrial agglomeration on income inequality from a spatial perspective.). There are few literatures on “Spatial effect of industrial agglomeration on income inequality”, thus, this is one of reasons that this paper investigates the impact of industrial agglomeration on income inequality from a spatial perspective. Regarding the robustness test, according to the reviewer’s suggestions, we have deleted “service industry agglomeration” and added manufacturing employment density to replace the independent variable. This paper uses maximum likelihood estimation (MLE) and the fixed effects to weaken the endogeneity problem. Furthermore, thanks for the reminding of the reviewer, we have added the one-period lagged manufacturing industrial agglomeration instead of the independent variable in robustness test to alleviate the potential endogeneity problem based on the research of Zhang et al. (2022). The revised robustness test is shown in Table 6. To make the description of variables clear and detailed, we have revised the table of variable description as follows. Some control variables’ maximums and minimums are very different, this study takes the logarithm of these variables. In Table 2, it can be seen that all variables’ standard deviations are small to meet standardization to a certain extend.

Point 4: The paper is not innovative enough, and the research conclusions are not novel enough.

Reply 4: Thanks for your valuable comments. This paper mainly has the following innovations: Firstly, this paper presents an up-to-date portrait of the regional dimensions of income inequality across the country since the 21st century, which has been rarely depicted in the past. Secondly, most existing studies only focus on the linear impact of industrial agglomeration on income inequality, but the non-linear changes are not detailed enough. Industrial agglomeration affects income inequality through the agglomeration effect and crowding effect. It is easy to ignore the impact of the dynamic changes of the two effects on income inequality in the process of industrial agglomeration, which leads most studies to focus only on the linear relationship of the two variables. Therefore, this paper empirically investigates whether industrial agglomeration has a nonlinear relationship with income inequality. Thirdly, most previous studies have used traditional panel data analysis, ignoring the spatial correlation and spatial spillover effects of regional income inequality and industrial agglomeration. Thus, this paper uses global Moran’s I and local Moran’s I to test the spatial correlation of industrial agglomeration and income inequality respectively, then adopts the spatial Durbin model (SDM) to study the effect of industrial agglomeration on income inequality in China from 2003 to 2020. The research conclusions are based on employing the spatial models which are different from those of previous studies. Finally, based on the reviewer’s comments, we will make further efforts to optimize this paper.

We really appreciate for editors and reviewers’ warm work, and hope that the corrections will meet with approval. Once again, thank you very much for your comments and suggestions.

Sincerely,

Suhua Zhang

zhangsuhua1023@gmail.com

---

## [Decision Letter · Decision Letter 2]

15 Jun 2023

Exploring the effect of industrial agglomeration on income inequality in China

PONE-D-22-32289R2

Dear Dr. Zhang,

We’re pleased to inform you that your manuscript has been judged scientifically suitable for publication and will be formally accepted for publication once it meets all outstanding technical requirements.

Kind regards,

Fuyou Guo, (Ph.D.

Academic Editor

PLOS ONE

Additional Editor Comments (optional):

All comments have been addressed.

Reviewers' comments:

Reviewer's Responses to Questions

**Comments to the Author**

1. If the authors have adequately addressed your comments raised in a previous round of review and you feel that this manuscript is now acceptable for publication, you may indicate that here to bypass the “Comments to the Author” section, enter your conflict of interest statement in the “Confidential to Editor” section, and submit your "Accept" recommendation.

Reviewer #2: All comments have been addressed

Reviewer #4: (No Response)

2. Is the manuscript technically sound, and do the data support the conclusions?

Reviewer #2: Yes

Reviewer #4: Partly

3. Has the statistical analysis been performed appropriately and rigorously? 

Reviewer #2: Yes

Reviewer #4: No

4. Have the authors made all data underlying the findings in their manuscript fully available?

Reviewer #2: Yes

Reviewer #4: Yes

5. Is the manuscript presented in an intelligible fashion and written in standard English?

Reviewer #2: Yes

Reviewer #4: Yes

6. Review Comments to the Author

Reviewer #2: (No Response)

Reviewer #4: (No Response)

7. PLOS authors have the option to publish the peer review history of their article (what does this mean?). If published, this will include your full peer review and any attached files.

Reviewer #2: No

Reviewer #4: No

---

## [Editor Report · Acceptance letter]

19 Jun 2023

PONE-D-22-32289R2 

Exploring the effect of industrial agglomeration on income inequality in China 

Dear Dr. Zhang:

I'm pleased to inform you that your manuscript has been deemed suitable for publication in PLOS ONE. Congratulations! Your manuscript is now with our production department. 

Kind regards, 

on behalf of

Associate professor Fuyou Guo 

Academic Editor

PLOS ONE